# High prevalence of atypical virulotype and genetically diverse background among *Pseudomonas aeruginosa* isolates from a referral hospital in the Brazilian Amazon

Yan Corrêa Rodrigues[1]*, Ismari Perini Furlaneto[2], Arthur Henrique Pinto Maciel[3], Ana Judith Pires Garcia Quaresma[3], Eliseth Costa Oliveira de Matos[4], Marília Lima Conceição[1], Marcelo Cleyton da Silva Vieira[3], Giulia Leão da Cunha Brabo[3], Edilene do Socorro Nascimento Falcão Sarges[1], Luana Nepomuceno Godim Costa Lima[1,3], Karla Valéria Batista Lima[1,3]*

**1** Programa de Pós-graduação em Biologia Parasitária na Amazônia, Centro de Ciências Biológicas e da Saúde, Universidade do Estado do Pará (UEPA), Belém, Pará, Brazil, **2** Programa de Pós-graduação em Educação em Saúde, Centro Universitário do Pará (CESUPA), Belém, Pará Brazil, **3** Laboratório de Biologia Molecular, Seção de Bacteriologia e Micologia, Instituto Evandro Chagas (IEC), Ministério da Saúde, Ananindeua, Pará, Brazil, **4** Departamento de Patologia, Centro de Ciências Biológicas e da Saúde, Universidade do Estado do Pará (UEPA), Belém, Pará, Brazil

* yan.13@hotmail.com (YCR); karlalima@iec.gov.br (KVBL)

## Abstract

*Pseudomonas aeruginosa* is an opportunistic pathogen causing different types of infections, particularly in intensive care unit patients. Characteristics that favor its persistence artificial environments are related to its high adaptability, wide arsenal of virulence factors and resistance to several antimicrobial classes. Among the several virulence determinants, T3SS stands as the most important due to the clinical impact of *exoS* and *exoU* genes in patient's outcome. The molecular characterization of *P. aeruginosa* isolates helps in the comprehension of transmission dynamics and enhance knowledge of virulence and resistance roles in infection process. In the present study, we investigated virulence and resistance properties and the genetic background of *P. aeruginosa* isolated from ICUs patients at a referral hospital in Brazilian Amazon. A total of 54 *P. aeruginosa* isolates were characterized by detecting 19 virulence-related genes, antimicrobial susceptibility testing, molecular detection of β-lactamase-encoding genes and genotyping by MLST and rep-PCR. Our findings showed high prevalence of virulence-related markers, where 53.7% of the isolates presented at least 17 genes among the 19 investigated ($P = 0.01$). The rare $exoS^+/exoU^+$ cytotoxic virulotype was detected in 55.6% of isolates. Antimicrobial susceptibility testing revealed percentages of antibiotic resistance above 50% to carbapenems, cephalosporins and fluoroquinolones associated to MDR/XDR isolates. Isolates harboring both $bla_{SPM-1}$ and $bla_{OXA}$ genes were also detected. Genotyping methods demonstrated a wide genetic diversity of strains spread among the different intensive care units, circulation of international MDR/XDR high-risk clones (ST111, ST235, ST244 and ST277) and emergence of seven novel MLST lineages. Finally, our findings highlight the circulation of strains with high virulence potential and resistance to antimicrobials and may be useful on comprehension of pathogenicity process,

**Data Availability Statement:** All relevant data are within the manuscript and its Supporting Information files.

**Funding:** This research was supported by funding from Fundação de Amparo à Pesquisa do Pará/ Universidade do Estado do Pará (FAPESPA/UEPA) [Cooperation grant N°004/2019], Programa Institucional de Bolsas de Iniciação Científica/ Instituto Evandro Chagas - Conselho Nacional de Desenvolvimento Científico e Tecnológico (PIBIC/ IEC-CNPq) and Instituto Evandro Chagas/ Ministério da Saúde/Secretaria de Vigilância em Saúde (IEC/MS/SVS). The funders had no role in study design, data collection and analysis, decision to publish, or preparation of the manuscript.

**Competing interests:** The authors declare no potential conflict of interests regarding the publication of this study.

treatment guidance and establishment of strategies to control the spread of epidemic *P. aeruginosa* strains.

## Introduction

*Pseudomonas aeruginosa* is an opportunistic pathogen, causing a wide spectrum of infections, including acute and chronic respiratory tract infections (RTIs) and bloodstream infections (BSIs) as the main infection sites [1–4]. It is estimated that *P. aeruginosa* is responsible to about 10% of all hospital-acquired-infections worldwide and associated to outbreaks in adult, pediatric and neonatal intensive care units (ICUs) due to the spread of multi-drug resistant (MDR) or extensively drug-resistant (XDR) and highly virulent *P. aeruginosa* strains, negatively impacting morbidity, mortality, length of stay and treatment costs of patients [5–7]. *P. aeruginosa* high adaptability, resistance to several antimicrobial classes and secretion of various virulence factors allows its persistence in artificial settings of high selective pressure, such as hospitals [8–10].

Pathogenicity in *P. aeruginosa* is multifactorial, relying on the regulation of virulence-related genes and expression of their respective factors, including adhesins, exotoxins, proteases and pigments. Virulence products may be passively secreted from bacterial cell or actively secreted via secretion systems, such as type I secretion system (T1SS), type II secretion system (T2SS) and type III secretion system (T3SS) [11–13]. T3SS is the most important and well characterized virulence determinant in *P. aeruginosa*, which allows the translocation of effector cytotoxic proteins into the host cell via a nano-syringe. So far, only four effector exotoxins encoded by the *exoS*, *exoU*, *exoT* and *exoY* genes have been identified, which are variably present and expressed by *P. aeruginosa* strains. ExoS and ExoU are associated to invasive and cytotoxic phenotypes, respectively, and rarely concomitant detected, while ExoT and ExoY demonstrate few cytotoxic effects and are encoded by most of strains [8, 14–17]. Furthermore, survey of cytotoxic/*exoU*$^+$ virulotype has been highly recommended due to the impact of this exotoxin in patient's mortality, especially in isolates from high risk settings, such as ICUs [18, 19].

The increasing trend of resistance to anti-pseudomonal antibiotics, especially carbapenems, aminoglycosides and fluoroquinolones has contributed to the emergence of *P. aeruginosa* MDR/XDR strains, posing a challenge to infection treatment [20–22]. Among the several resistance mechanisms in *P. aeruginosa*, the role of transferable resistance determinants is overwhelming, particularly those encoding classes A and D extended-spectrum β-lactamases (ESBLs) and class B metallo-β-lactamases (MBLs) [23–26].

Molecular typing of in-hospital isolates of *P. aeruginosa* contribute to the understanding of transmission and infection dynamics by strains exhibiting high degree of virulence and resistance, allowing the determination of genetic relationships between isolates, outbreaks identification, population structure studies and therapeutic management [27–30]. In spite of the polyclonal population structure of *P. aeruginosa*, several studies have shown a globally spread of strains denominated as high-risk clones (HRC), which are associated with most MDR/XDR isolates and specific virulence markers. Thus, characterization of *P. aeruginosa* isolates must be performed combining clonality evaluation with methods that can identify specific virulence and resistance determinants [19, 23, 31, 32].

The molecular characterization of local *P. aeruginosa* isolates is one of the first steps to understand the epidemiology of the pathogen at local and global level, helping to establish

surveillance strategies, reduce risk of outbreaks and enhance knowledge of virulence and resistance roles in infection process. Nonetheless, comprehensive epidemiological investigations on molecular level are still scarce, especially ones reporting data from Brazilian territory. In the present study we report in-depth data into the virulence, resistance properties and genetic diversity of in-patient ICU isolates from a referral hospital in the Brazilian Amazon region.

## Material and methods

### Bacterial isolates

In this retrospective cross-sectional study, a total of 54 non-repetitive *P. aeruginosa* isolates stored in the Bacteriology and Mycology Section of the Evandro Chagas Institute were evaluated. Isolates were previously recovered of various clinical sources of patients hospitalized in different ICUs at a referral hospital, from January 2010 to June 2013 (S1 Table). Identification of isolates was performed on automated VITEK-2 system (bioMérieux, Marcy l'Etoile, France) and complemented by phenotypical and biochemical assays such as observation of colony morphology, Gram staining, oxidase test, absence of carbohydrate fermentation on triple sugar iron agar, oxidative behavior on Hugh Leifson medium and the cytochrome oxidase reaction.

### DNA extraction and molecular detection of virulence-related genes

Genomic DNA was obtained from a single colony of overnight cultures for each *P. aeruginosa* isolate using UltraClean™ Microbial DNA Isolation kit (MoBio Laboratories, Carlsbad, CA, USA), following manufacturer's recommendations and quantified using the Picodrop PICO100 spectrophotometer (Picodrop Limited, Hinxton, UK). Adjusted DNA concentrations between 25—50ng/μl were used for all subsequent molecular assays.

The detection of 19 virulence-related markers was performed by PCR: adhesion (*algU* and *algD*), T1SS (*aprA*), T2SS (*lasA*, *lasB* and *toxA*), T3SS (*exoS*, *exoU*, *exoT* and *exoY*), oxidative stress (*phzI*, *phzII*, *phzM*, *phzS* and *phzH*) and quorum sensing (QS)/regulation (*lasI*, *lasR*, *rhlL* and *rhlR*), using previously described primers and reaction parameters with slightly modifications [33–38]. Each PCR mixture was prepared in a final volume of 25 μl, consisted of 1x PCR buffer (Qiagen, Hilden, Germany), 1x Q-solution (Qiagen, Hilden, Germany), 2 mM MgCl$_2$ (Qiagen, Hilden, Germany), 200 μM each dNTP (Invitrogen™, São Paulo, Brazil), 0,5 μM for each primer, 1U of HotStart Taq DNA polymerase and DNA template. Amplifications were performed on Veriti thermocycler (Applied Biosystems, Foster City, CA, USA), under the following conditions for all genes: 95˚C for 15 minutes; followed by 35 cycles at 95˚C for 1 minute, 60˚C for 45 seconds, 72˚C for 1 minute and a final extension step at 72˚C for 7 minutes. PCR products were subjected to electrophoresis on 1.5% agarose gel and amplicons visualized under ultraviolet light. *P. aeruginosa* ATCC 27853 and PA14 strains were used as positive control.

### Antimicrobial susceptibility assays and definition of resistance phenotypes

Antimicrobial susceptibility testing was determined as the Minimum Inhibitory Concentration (MICs) by the broth microdilution method for six antimicrobial classes: cephalosporins (cefepime – FEP), carbapenems (imipenem – IMP and meropenem – MER), penicilins + beta-lactamase inhibitor (piperacillin+tazobactam – TZP), aminoglycosides (gentamicin – GN), fluoroquinolones (ciprofloxacin – CIP) and polypeptides (polymyxin B – POL). All assays were performed in accordance to the Clinical and Laboratory Standards Institute guidelines and breakpoints [39] and isolates were classified as sensitive (S), intermediate (I) and

resistant (R). Reference strains *Escherichia coli* ATCC 25922 and *P. aeruginosa* ATCC 27853 were used as quality control. For statistical analysis, isolates exhibiting resistant and intermediate susceptibility phenotypes were grouped as 'non-susceptible'. Isolates were phenotypically classified as MultiS if susceptible to all tested antimicrobial classes; Moderately resistant (ModR) if resistant to $\geq 1$ drug in $< 3$ antimicrobial classes; MDR if resistant to $\geq 1$ drug in $\geq 3$ antimicrobial classes and XDR if non-susceptible to 1 agent in all but $\leq 2$ tested antimicrobial classes, according to previously described criteria [31, 40].

## Molecular detection of antimicrobial resistance markers

The detection of β-lactamase-encoding genes, including ESBLs ($bla_{CTX-M1}$, $bla_{OXA-2}$, $bla_{OXA-10}$) and carbapenemases ($bla_{IMP-1}$, $bla_{IMP-2}$, $bla_{VIM-1}$, $bla_{VIM-2}$, $bla_{SPM-1}$, $bla_{NDM-1}$ and $bla_{KPC-1}$) was performed by PCR on Veriti thermocycler (Applied Biosystem, Foster City, CA, USA) as previously described [41–47]. PCR products were subjected to electrophoresis on 1.5% agarose gel and amplicons visualized under ultraviolet light.

## Molecular typing based on MLST

MLST genotyping was performed according to protocol described by Curran et al. [48]. The seven housekeeping genes (*acsA*, *aroE*, *guaA*, *mutL*, *nuoD*, *ppsA*, and *trpE*) were amplified by PCR in a Veriti thermocycler (Applied Biosystems, Foster City, CA, USA). Reaction products were bidirectionally sequenced using Big Dye Terminator v3.1 chemistry on ABI Prism 3100 Genetic Analyzer (Applied Biosystems, Foster City, CA, USA). Obtained results were compared to available data at PubMLST database (http://pubmlst.org/paeruginosa) to determine allelic profiles and sequence types (STs). Novel alleles and STs were submitted to PubMLST database for validation. PHYLOViZ 2.0 platform was used for data management and analysis of clonal complexes (CCs), which were defined by related ST clusters exhibiting variation in a single locus (single locus variants—SLV) or in two loci (double locus variants —DLV) [49].

## Molecular typing based on rep-PCR

The genetic relatedness of isolates was investigated by semi-automated rep-PCR on Diversi-Lab™ Strain Typing System (bioMérieux, Marcy-L'Étoile, France) using the DiversiLab *P. aeruginosa* kit (Bacterial Barcodes, bioMérieux, Marcy-L'Étoile, France), according to the manufacturer's instructions. Fingerprints were obtained by electrophoresis using microfluidic lab-on-a-chip on Agilent 2100 Bioanalyzer equipment (Agilent Technologies, Palo Alto, CA, USA) and analysis performed with DiversiLab on-line software (v 3.4) applying the Pearson correlation coefficient. Isolates were classified as in the same clonal group (genotypically indistinguishable) if the similarity was $\geq 97\%$, and as unique pattern if the similarity was $< 97\%$.

## Statistical analysis

The G-test of independence or Fisher's exact test was applied to verify the association between resistance, virulence markers, isolation sites, clonal groups and STs; standardized residuals and adjusted residuals was used as a post-hoc test after a statistically significant G-test of independence. The distribution of virulence-related genes among isolates was verified by the Lilliefors test. Values of $P \leq 0.05$ were considered statistically significant. All analyzes were performed using the statistical software BioEstat® 5.4 [50].

### Ethics statement

No samples were collected for this study. Only stored samples were included without any contact and possibility of identifying the respective patients. Prior sampling and the present study were conducted in accordance with Helsinki Declaration and the Brazilian National Health Council [51] and with approval of the ethics committee at Fundação Santa Casa de Misericórdia do Pará (referral hospital under study) (N° CAAE 0086.0.440.000–10).

## Results

### Bacterial isolates and distribution of virulence-related genes

This study evaluated 54 *P. aeruginosa* isolates distributed in the adult ICU (AICU, 28/54–51.8%), followed by pediatric ICU (PICU, 14/54–26.0%) and neonatal ICU (NICU, 12/54–22.2%). Regarding clinical origin, isolates were obtained from RTI (23/54–42.6%), BSI (14/54–25.9%) and others various sources (17/54–31.5%), including catheter (n = 6), rectal swab (n = 3), surgical wound (n = 3), ocular secretion (n = 2), gastric secretion (n = 1), urethral secretion (n = 1), urine (n = 1) (S1 Table).

Evaluated isolates harbored at least five virulence-related genes and seven isolates harbored all 19 investigated genes. In addition, a heterogeneous distribution of genes was observed, where 53.7% of the isolates presented at least 17 genes among the 19 investigated ($P = 0.01$). The lowest detection frequencies were observed for *lasA* (21/54–38.9%) and *algU* (25/54–46.3%) genes, while *lasB*, *exoS*, *rhlL* and *rhlR* genes were detected in all isolates (54/54–100%) (S1 Fig and S1 Table). Definition of virulotypes was based on the detection of T3SS *exoS/exoU* genes, where it was observed the presence of invasive/cytotoxic ($exoS^+/exoU^+$; 30/54–55.6%) and invasive ($exoS^+/exoU^-$; 24/54–44.4%) virulotypes. $exoS^+/exoU^+$ virulotype and *phzH* gene were detected with significantly lower frequency at NICU ($P = 0.0461$ and $p = 0.0491$, respectively), and among isolates from BSI ($P = 0.0027$ and $P = 0.0244$, respectively), as well as *exoY* gene ($P = 0.0435$) (Tables 1 and 2).

### Antimicrobial susceptibility features

Antimicrobial susceptibility testing revealed that isolates were mainly non-susceptible to carbapenems (IMP and MER; 36/54–66.7%), followed by FEP (27/54–50.0%) and CIP (27/54–50.0%), GN (24/54–44.4%) and TZP (17/54–31.5%). All isolates were susceptible to POL (S1 Table). According to phenotypical classification, isolates were classified as ModR (23/54–42.6%), followed by XDR (16/54–29.6%), MultiS (9/54–16.7%) and MDR (6/54–11.1%). MDR/XDR isolates were significantly predominant in the AICU (Table 3) ($P = 0.0003$). There was no significant association between the presence of T3SS virulotypes and antimicrobial resistance (Table 4), where 33.3% (18/30) $exoU^+$ were classified as MultiS or ModR.

Regarding molecular survey of antimicrobial resistance genes, 20.4% (11/54) of *P. aeruginosa* isolates harbored $bla_{CTX-M1}$, followed by $bla_{SPM-1}$ (5/54–9.2%), $bla_{OXA-2}$ (3/54–5.5%) and $bla_{OXA-10}$ (3/54–5.5%). Two isolates harbored both $bla_{SPM-1}$ and $bla_{OXA-2}$ and $bla_{SPM-1}$ and $bla_{OXA-10}$ genes. One isolate harbored $bla_{OXA-2}$ gene and one isolate harbored $bla_{OXA-10}$. Four isolates harboring $bla_{SPM-1}$ genes and seven isolates harboring $bla_{CTX-M1}$ were classified as XDR. The $bla_{IMP}$, $bla_{VIM}$, $bla_{NDM}$ and $bla_{KPC}$ genes were not detected.

### Molecular typing based on MLST

MLST genotyping revealed a highly diverse genetic background with the presence of 22 different STs, including seven novel STs (ST2524, ST2541, ST2552, ST2554, ST2555, ST 2556 and

**Table 1. Distribution of virulence-related genes according to patient's ICUs.**

| Related function | Gene | AICU $n = 28$ | PICU $n = 14$ | NICU $n = 12$ | *P*-value* |
|---|---|---|---|---|---|
| Adhesion | *algU* | 12 | 7 | 6 | 0.8758 |
| | *algD* | 28 | 13 | 12 | 0.4601 |
| T1SS | *aprA* | 21 | 10 | 6 | 0.3204 |
| T2SS | *lasA* | 9 | 7 | 5 | 0.5379 |
| | *lasB* | 28 | 14 | 12 | 1.000 |
| | *toxA* | 23 | 13 | 11 | 0.9329 |
| T3SS | *exoS$^+$/ exoU$^+$* | 17 | 10 | 3[a] | 0.0461* |
| | *exoS$^+$/ exoU$^-$* | 11 | 4 | 9 | |
| | *exoT* | 21 | 12 | 6 | 0.1371 |
| | *exoY* | 23 | 12 | 6 | 0.0863 |
| Oxidative stress | *phzI* | 27 | 13 | 11 | 0.8346 |
| | *phzII* | 21 | 11 | 5 | 0.0959 |
| | *phzM* | 22 | 12 | 7 | 0.2775 |
| | *phzS* | 20 | 12 | 5 | 0.0581 |
| | *phzH* | 21 | 12 | 5[a] | 0.0491* |
| QS/regulation | *lasI* | 26 | 13 | 12 | 0.5386 |
| | *lasR* | 26 | 12 | 12 | 0.3334 |
| | *rhlL* | 28 | 14 | 12 | 1.000 |
| | *rhlR* | 28 | 14 | 12 | 1.000 |

T1SS, type I secretion system; T2SS, type II secretion system; T3SS, type III secretion system; QS, quorum sensing; AICU, adult intensive care unit; PICU, pediatric intensive care unit; NICU, neonatal intensive care unit.

* *P* values were calculated using the G-test of independence.

[a] Frequency lower than expected at random.

ST2603) and 15 STs previously reported (ST111, ST170, ST235, ST244, ST277, ST360, ST463, ST500, ST508, ST1076, ST1197, ST1284, ST1655, ST2100 and ST2437). Novel allele 218/*aroE* was associated to ST2603. Genetic relationship analysis demonstrated the presence founders STs of four HRC CC, including CC/ST111 (2/22–9.0%), CC/ST235 (5/22–22.7%), CC/ST244 (9/22–40.9%) and CC/ST277 (6/22–27.2%). Among the seven novel STs, ST 2554 emerged as DLV of CC/ST274, which is a HRC; ST 2555 as SLV of ST 316 and the other five STs (ST2524, ST2541, ST2552, ST 2556 and ST2603) were singletons.

Among the isolates, 40.7% (22/54) were associated to HRC and 59.3% (32/54) to non-HRC. HRC were significantly predominant in the AICU ($P = 0.0100$) (Table 5), especially ST244 (8/28–28.5%). Although no significant association between MLST genotypes and T3SS virulotypes was observed, 50% (11/22) of HRC presented *exoS$^+$/exoU$^+$* virulotype ($P = 0.5820$) (Table 5). Non-susceptibility to FEP, IMP, MER, GN, CIP ($P < 0.001$) and TZP ($P = 0.0198$) was predominant among HRC strains (Table 5). All 16 XDR isolates were associated to the detected HRC ($P < 0.0001$) (Table 5) and only one MDR isolate belonged to non-HCR (ST2524); whereas all ModR isolates were associated to non-HRC ($P < 0.0001$) (Table 5) and only one MultiS isolate belonged to HCR (ST244) (Fig 1). Five isolates harboring *bla*$_{SPM-1}$, *bla*OXA$_{-2}$ and *bla*$_{OXA-10}$ genes belonged to HRC ST277 and one isolate harboring only *bla*OXA$_{-2}$ was associated to non-HRC ST508; while isolates harboring *bla*$_{CTX-M1}$ belonged to HRC ST111, ST244, ST235 (10/11) and to non-HRC ST1284 (1/11).

**Table 2. Distribution of virulence-related genes according to the *P. aeruginosa* isolates clinical sources.**

| Related function | Gene | RTI n = 23 | BSI n = 14 | Other n = 17 | *P*-value* |
|---|---|---|---|---|---|
| Adhesion | *algU* | 12 | 6 | 7 | 0.7617 |
| | *algD* | 23 | 14 | 16 | 0.5003 |
| T1SS | *aprA* | 15 | 7 | 15 | 0.0618 |
| T2SS | *lasA* | 7 | 5 | 9 | 0.3569 |
| | *lasB* | 23 | 14 | 17 | 1.000 |
| | *toxA* | 21 | 13 | 15 | 0.9132 |
| T3SS | *exoS*+/ *exoU*+ | 13 | 3[a] | 14 | 0.0027* |
| | *exoS*+/ *exoU*- | 10 | 11 | 3 | |
| | *exoT* | 17 | 7 | 15 | 0.0670 |
| | *exoY* | 19 | 7[a] | 15 | 0.0435* |
| Oxidative stress | *phzI* | 23 | 13 | 15 | 0.2276 |
| | *phzII* | 17 | 7 | 13 | 0.2489 |
| | *phzM* | 18 | 8 | 15 | 0.1449 |
| | *phzS* | 17 | 6 | 14 | 0.0596 |
| | *phzH* | 17 | 6[a] | 15 | 0.0244* |
| QS/regulation | *lasI* | 21 | 13 | 17 | 0.3815 |
| | *lasR* | 21 | 13 | 16 | 0.9518 |
| | *rhlL* | 23 | 14 | 17 | 1.000 |
| | *rhlR* | 23 | 14 | 17 | 1.000 |

T1SS, type I secretion system; T2SS, type II secretion system; T3SS, type III secretion system; QS, quorum sensing; RTI, respiratory tract infection; BSI, bloodstream infection; Other: catheter (n = 6), rectal swab (n = 3), surgical wound (n = 3), ocular secretion (n = 2), gastric secretion (n = 1), urethral secretion (n = 1), urine (n = 1).

* *P* values were calculated using the G-test of independence.

[a] Frequency lower than expected at random.

## Molecular typing based on rep-PCR

Genotyping by rep-PCR on DiversiLab™ System revealed 36 distinct fingerprint patterns, of which 26 were unique patterns and ten clonal groups comprised of two to eight isolates (Fig 2). Strains associated to clonal groups were found circulating in different ICUs, however, no statistically significant association was observed between clonality and ICUs (*P* = 0.5063) (Table 6). The frequency of virulotypes was similar between the different clonal patterns defined by DiversiLab™ (*P* = 1.000) (Table 6). MDR/XDR isolates were found in higher

**Table 3. Distribution of *P. aeruginosa* isolates according to susceptibility phenotype among the ICUs in the referral hospital.**

| | MultiS/ModR | MDR/XDR | *P*-value* |
|---|---|---|---|
| **ICU** | | | |
| AICU | 11 | 17 | |
| PICU | 9 | 5 | 0.0003 |
| NICU | 12 | 0 | |

ICU, intensive care unit; AICU, adult intensive care unit; PICU, pediatric intensive care unit; NICU, neonatal intensive care unit; MultiS, susceptible to all tested antibiotics; ModR, moderately resistant; MDR, multi-drug resistant; XDR, extensively drug-resistant.

* *P* value was calculated using the G-test of independence.

**Table 4. Distribution of T3SS virulotypes according to antimicrobial susceptibility of *P. aeruginosa* isolates.**

| | *exoS*⁺/*exoU*⁺ $n = 30$ | *exoS*⁺/*exoU*⁻ $n = 24$ | *P*-value* |
|---|---|---|---|
| **Antimicrobial resistance** | | | |
| FEP | 15 | 12 | 1.000 |
| IMP | 21 | 15 | 0.5770 |
| MER | 15 | 11 | 0.7905 |
| TZP | 9 | 8 | 1.000 |
| GN | 13 | 11 | 1.000 |
| CIP | 14 | 13 | 0.7846 |
| **Susceptibility phenotype** | | | |
| MultiS | 3 | 6 | 0.2702 |
| ModR | 15 | 8 | 0.2738 |
| MDR | 5 | 1 | 0.2100 |
| XDR | 7 | 9 | 0.3695 |
| MDR+XDR | 12 | 10 | 1.000 |

FEP, cefepime; IMP, imipenem; MER, meropenem; TZP, piperacillin+tazobactam; GN, gentamicin; CIP, ciprofloxacin; POL, polymyxin B; MultiS, susceptible to all tested antibiotics; ModR, moderately resistant; MDR, multi-drug resistant; XDR, extensively drug-resistant.

* *P* values were calculated using the Fisher's exact test.

proportion among clonal groups strains ($P = 0.0141$) (Table 6). Isolates harboring $bla_{SPM-1}$, $bla$OXA$_{-2}$ and $bla_{OXA-10}$ genes were related to clonal groups A, B and C/ST277, while $bla_{CTX-M1}$ gene was distributed among clonal groups (D, E, H and I) and unrelated strains (Fig 2). Genetic diversity observed through MLST genotyping presented a good association with fingerprint diversity found by rep-PCR, of which seven clonal groups were associated to HRC, but without reaching significant association ($P = 0.0573$) (Table 6).

## Discussion

*P. aeruginosa* is currently one of the most relevant opportunistic pathogens causing acute infections, particularly among patients admitted to ICUs [5–6]. The occurrence of infection episodes and mortality in ICUs is significantly higher compared to other hospital wards; this scenario is justified by the complexity of the patients' clinical condition, such as immunosuppression and pre-existing diseases, use of invasive diagnostic and therapeutic maneuvers, as well as, the spread of epidemic and highly virulent strains of *P. aeruginosa* [2, 4, 23, 32]. *P. aeruginosa* isolates included in our study were mainly recovered from RTI (42.6%) and BSI (25.9%) of patients admitted to different ICUs, similarly as previously reported infections frequencies, which varied between 9% to 35% in adult ICUs, reaching 62% in neonatal and pediatric ICUs and with RTI and BSI as major clinical sources [2, 3, 4, 6, 52, 53].

Although considered an opportunistic pathogen, the myriad of virulence products of *P. aeruginosa* may severally impact patients' clinical condition. The combination of multiple genes and virulence factors tends to affect pathogenesis and determine the outcome of an infectious process, and depending on the location and type of infection, the importance and role of a given factor may be different [10, 38, 54]. The detection of virulence-related markers revealed a high prevalence of genes, with detection of at least five genes per isolate and 17 of the 19 genes investigated in more than 50% of the isolates ($P = 0.01$), demonstrating a high virulence potential of tested strains (S1 Fig). Although our findings are consistent with several other

**Table 5. Distribution of HRC and non-HRC according to ICUs, T3SS virulotypes and antimicrobial susceptibility of *P. aeruginosa* isolates.**

| | HRC *n* = 22 | non-HRC *n* = 32 | *P*-value* |
|---|---|---|---|
| **ICU** | | | |
| AICU | 16[a] | 12[b] | 0.0100 |
| PICU | 5 | 9 | |
| NICU | 1[b] | 11[a] | |
| **T3SS Virulotype** | | | |
| *exoS+/exoU+* | 11 | 19 | 0.5820 |
| *exoS+/exoU-* | 11 | 13 | |
| **Antimicrobial resistance** | | | |
| FEP | 18 | 9 | 0.0002 |
| IMP | 20 | 16 | 0.0027 |
| MER | 17 | 9 | 0.0007 |
| TZP | 11 | 6 | 0.0198 |
| GN | 20 | 4 | < 0.0001 |
| CIP | 21 | 6 | < 0.0001 |
| **Susceptibility phenotype** | | | |
| MultiS | 1 | 8 | 0.0676 |
| ModR | 0 | 23 | < 0.0001 |
| MultiS+ModR | 1 | 31 | < 0.0001 |
| MDR | 5 | 1 | 0.0706 |
| XDR | 16 | 0 | < 0.0001 |
| MDR+XDR | 21 | 1 | < 0.0001 |

HRC, high-risk clones; ICU, intensive care unit; AICU, adult intensive care unit; PICU, pediatric intensive care unit; NICU, neonatal intensive care unit; T3SS, type III secretion system; FEP, cefepime; IMP, imipenem; MER, meropenem; TZP, piperacillin+tazobactam; GN, gentamicin; CIP, ciprofloxacin; POL, polymyxin B; MultiS, susceptible to all tested antibiotics; ModR, moderately resistant; MDR, multi-drug resistant; XDR, extensively drug-resistant.

* *P* values were calculated using the G-test of independence or Fisher's exact test.

[a] Frequency higher than expected at random.

[b] Frequency lower than expected at random.

reports that indicates a high prevalence and conserved nature of the *P. aeruginosa* genome regarding the presence of virulence-related genes, variations in virulence patterns are observed in isolates from different geographic areas and settings, highlighting the need to investigate different markers among distinct *P. aeruginosa* populations [1, 13, 23, 35, 55, 56].

The *algD* gene was detected in 98.1% of isolates, while absence of *algU* was noted in more than 53.0%. These findings differ from those reported by Hassuna et al (2020) [1] and Fazeli et al (2014) [57], who observed higher frequencies for *algU* and lower frequencies for *algD*, respectively. As both genes encode fundamental proteins for alginate biosynthesis, functioning as one of its main adhesins and mainly found in the respiratory tract of patients with acute infections and cystic fibrosis (CF), the low frequency of the *algU* may be associated to the high frequency of non-RTI isolates (57.4%) and also indicate deficient alginate production by most of tested isolates [10, 58].

The high prevalence of the *aprA* gene is in line with other studies [1, 13, 56, 59], highlighting the importance of zinc-metalloprotease secreted by T1SS, which has functions related to invasion, causing collagen and fibrinogen degradation in synergy with T2SS elastases, and

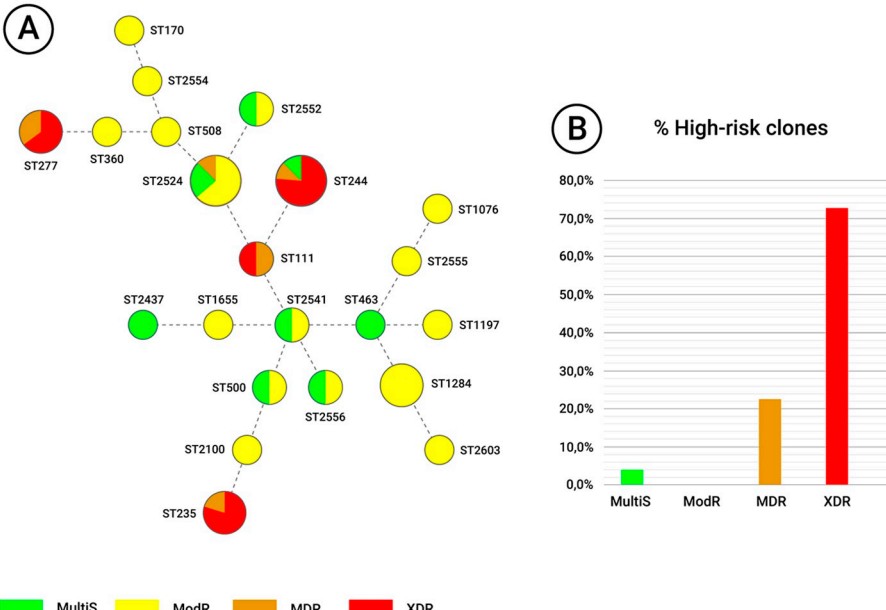

**Fig 1. Minimum spanning tree of the 54 *P. aeruginosa* isolates from the referral hospital in the Brazilian Amazon genotyped by MLST.** (A) Each circle represents a different ST and the size of the circle is proportional to the number of isolates related to the respective ST. Each of the four susceptibility phenotypes is represented by a different color. Dotted lines represent multilocus variants. (B) The graph shows the percentages of HRC associated with a particular susceptibility phenotype.

evasion of the immune system through the inactivation of several cytokines, contributing to bacterial survival [60, 61]. Three proteases—two elastases (LasA and LasB) and an exotoxin (ExoA)—secreted by *P. aeruginosa* are constituents of T2SS. The detection frequencies of 87.0% for *toxA* and 100.0% for *lasB*, in addition to the absence *lasA* among more than 60% of the tested strains are in agreement with several other studies [1, 13, 56, 57, 62, 63]. Elastases act synergistically and have high elastolytic capacity, mainly in blood vessels and lung tissues, while ExoA inhibits host protein synthesis leading to cell death [64, 65]. Thus, our results reinforce the hypothesis of a significantly higher prevalence of *lasB* over *lasA* in *P. aeruginosa* clinical isolates [37, 66].

The phenazine compounds encoded by the *phzI*, *phzII*, *phzH*, *phzM* and *phzS* genes have siderophoric activity, cause an increase in oxidative stress and mitochondrial inactivation [10, 67]. Interestingly, Bradbury et al. [13] correlated a high prevalence of these genes to *P. aeruginosa* clinical isolates, disagreeing with the low prevalence reported among strains tested in this study, and mainly of *phzH* gene, which its absence was significant among isolates from BSI ($P = 0.0244$) at NICU ($P = 0.0491$) (Tables 1 and 2).

The QS genes were detected at frequencies above 92.0%, however, four isolates did not harbor *lasR* gene and three isolates *lasI* gene. The QS system in *P. aeruginosa* is coordinated by the *las* and *rhl* systems that function hierarchically for bacterial survival in the face of environmental changes, biofilm formation and control of several other virulence factors [10, 12]. Although a decrease in the ability to express virulence factors has been reported due to the absence and/or mutations in genes related to such QS, this does not necessarily compromise the ability to cause infection due to regulation mediated by the *rhl* system [12, 68, 69] Our results are in agreement with a study carried out by Senturk et al. [12] and Ruiz-Roldán et al. [56]; however, disagreeing with Aboushleib et al. [69], who reported gene deficiency in more than 60% in clinical isolates.

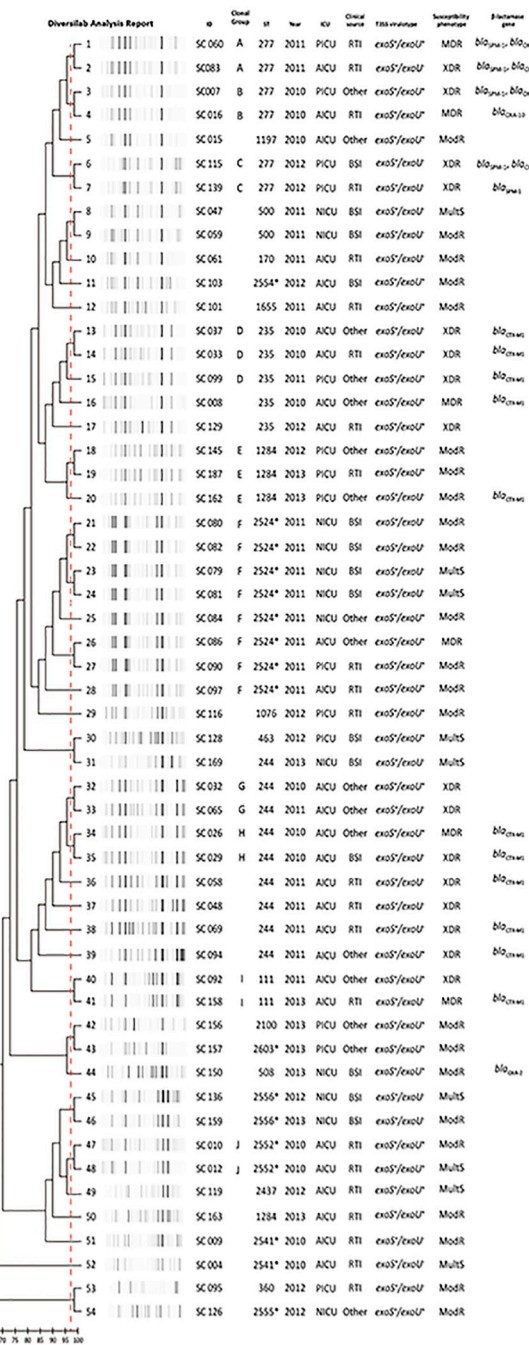

**Fig 2. Dendrogram of genetic similarity of the 54 *P. aeruginosa* isolates from the referral hospital in the Brazilian Amazon genotyped by rep-PCR in the DiversiLab™ system.** * Novel STs. ST, sequence type; ICU, intensive care unit; AICU, adult intensive care unit; PICU, pediatric intensive care unit; NICU, neonatal intensive care unit; RTI, respiratory tract infection; BSI, bloodstream infection; T3SS, type III secretion system; MultiS, susceptible to all tested antibiotics; ModR, moderately resistant; MDR, multi-drug resistant; XDR, extensively drug-resistant.

T3SS is the most important virulence marker in *P. aeruginosa*. The *exoS*, *exoU*, *exoT* and *exoY* genes encode toxins which are injected directly into the cytosol of the host cells and have functions related to anti-phagocytosis, necrotic and cytotoxic damage by a phospholipase, prevention of healing process and edema formation, respectively [10, 16, 54]. T3SS genes are

**Table 6. Distribution of rep-PCR patterns according ICUs, T3SS virulotypes and antimicrobial susceptibility of *P. aeruginosa* isolates.**

| | Clonal Group $n = 28$ | Unique patterns $n = 26$ | *P*-value* |
|---|---|---|---|
| **ICU** | | | |
| AICU | 14 | 14 | 0.5063 |
| PICU | 9 | 5 | |
| NICU | 5 | 7 | |
| **T3SS Virulotype** | | | |
| *exoS+/exoU+* | 16 | 14 | 1.000 |
| *exoS+/exoU-* | 12 | 12 | |
| **Susceptibility phenotype** | | | |
| MultiS | 3 | 6 | 0.2863 |
| ModR | 9 | 14 | 0.1683 |
| MultiS+ModR | 12 | 20 | 0.0141 |
| MDR | 5 | 1 | 0.1938 |
| XDR | 11 | 5 | 0.1410 |
| MDR+XDR | 16 | 6 | 0.0141 |
| **MLST** | | | |
| HRC | 15 | 13 | 0.0573 |
| non-HRC | 7 | 19 | |

AICU, adult intensive care unit; PICU, pediatric intensive care unit; NICU, neonatal intensive care unit; T3SS, type III secretion system; MultiS, susceptible to all tested antibiotics; ModR, moderately resistant; MDR, multi-drug resistant; XDR, extensively drug-resistant; HRC, high-risk clones.

* *P* values were calculated using the G-test of independence or Fisher's exact test.

variably present in isolates of *P. aeruginosa*, being the chromosomal genes *exoS*, *exoT* and *exoY* distributed among almost all strains [16, 18, 23]. T3SS *exoS*, *exoT* and *exoY* genes were detected in 100%, 72.2% and 75.9%, respectively. *exoS* results are in accordance with the hypothesis of a wider distribution of this gene due to its chromosomal nature. Nevertheless, the fact of in nearly 30% of our isolates *exoT* and *exoY* genes were not detected and a significant absence of *exoY* gene among BSI isolates demonstrate a contrary trend to the hypothesis of their universal distribution among *P. aeruginosa* strains, as well as, several studies that rarely report absence of these genes ($P = 0.0435$) (Table 2) [13, 15, 16, 55, 56, 70].

The *exoU* gene is found less frequently due to its presence being associated to the PAPI-2 in the accessory genome of *P. aeruginosa*, however, its presence and the secretion of its respective protein are markers of very highly cytotoxic phenotype and associated with several types of acute infections, antimicrobial resistance phenotype, polymicrobial infections and early mortality individuals [18, 31, 71, 72]. Moreover, *exoS/exoU* genes tend to be mutually exclusive, being observed a higher prevalence of the invasive virulotype ($exoS^+/exoU^-$) among *P. aeruginosa* strains worldwide [16, 19, 23, 59]. Interestingly, 55.6% of tested isolates presented the invasive/cytotoxic virulotype ($exoS^+/exoU^+$). Highlighting the rare occurrence of this phenomenon, there are only a few studies reporting a high proportion of this virulotype among *P. aeruginosa* hospital isolates [15, 35, 55]. We also observed a significant absence of $exoS^+/exoU^+$ virulotype among BSI isolates ($P = 0.0027$) and in the NICU ($P = 0.0461$), ward in which occurred the majority of BSI episodes among admitted patients (10/12–83.3%) (Tables 1 and 2), contradicting the important association established by several authors between the presence of $exoS^+/exoU^+$ virulotype among BSI isolates [18, 55, 73, 74]. Despite reports, the concomitant

presence of genes still is not well understood, considering: (I) absence of a direct relationship between the genes, where *exoS*, *exoT* and *exoY* are found in the core genome of *P. aeruginosa*, while *exoU* is found associated to the PAPI-2 in the accessory genome; (II) the impact on bacterial fitness due to the presence of both genes, which would stimulate the loss of one of them; and (III) the production of both toxins could generate a higher immune response, making the pathogen less capable of establishing infection due to its high immunogenicity [13, 16, 19, 23, 71].

Our findings revealed that more of 66% of tested isolates were resistant to carbapenems, followed by high resistance to cephalosporins, fluoroquinolones and aminoglycosides. Moreover, 40.7% of tested isolates were classified as MDR or XDR, which were significantly predominant in the AICU ($P$ = 0.0003) (Table 3) and associated to genetically related clonal groups ($P$ = 0.0141) (Table 6). For instance, our findings agree with recent data reported by Santos et al. [20], which also showed an increasing trend on resistance to carbapenems, aminoglycosides and fluoroquinolones associated to spread of *P. aeruginosa* MDR/XDR strains over 21 years' period in Rio de Janeiro, Brazil. Others studies conducted in Brazil and Latin America also revealed high resistance rates to cephalosporins, fluoroquinolones and aminoglycosides [75–78]. These findings emphasize the growing global trend on antibiotic resistance been reported by several studies and rises concern, especially in Latin America countries, where carbapenems still remains as the main choice for the treatment of infections by MDR *P. aeruginosa* [21, 22, 23, 79, 80]. A reduced number of transmissible resistance markers were detected among tested isolates, with $bla_{CTX-M1}$ (20.4%– 11/54) as the most frequent, followed by $bla_{SPM-1}$ (5/54–9.2%), $bla_{OXA-2}$ (3/54–5.5%) and $bla_{OXA-10}$ (3/54–5.5%). Our data demonstrate the predomination and wide spread of SPM-1 MDR/XDR strains in Brazilian hospitals and with the increasing transmission trend of ESBLs CTX-M and OXA variants genes worldwide, as previously reported [20, 23, 45, 76, 81, 82]. Finally, our results highlight the impact of horizontally acquired resistance mechanisms, mainly ESBLs and class B MBL, in the increasing resistance rates and its commonly association to MDR/XDR *P. aeruginosa* strains [18, 21, 23, 79].

The interaction between virulence and antibiotic susceptibility has been subject of several studies, suggesting that these relationships are antagonistic, since the presence of resistance mechanisms can determine a compromising biological cost for bacterial virulence and vice versa. In fact, the acquisition of resistance genes through the incorporation of mobile genetic elements in the bacterial chromosome can lead to the inactivation of genes involved in the production of some virulence factors and decreased cytotoxicity [23, 72, 74, 83]. Nevertheless, other authors suggest that $exoU^+$ virulotype is associated to resistance to fluoroquinolones and carbapenems, pointing out that the relationship between virulence and resistance may also occur in a synergistic sense, especially in high antibiotic pressure settings, such as ICUs [9, 13, 15, 16, 84]. In spite of that, our data do not suggest this association, as the presence of T3SS virulotypes was not related to any antimicrobial class or resistance phenotype (Table 4). Yet, 40% of $exoS^+/exoU^+$ isolates were associated to MDR/XDR phenotype, rising concern as a high prevalence of $exoS^+/exoU^+$ could facilitate the virulotype transmission to other MDR/XDR isolates, worsening the clinical scenario in the growing trend dissemination highly virulent MDR/XDR strains.

MLST revealed the presence of 22 different STs, including four different HRC (ST111, ST235, ST244 and ST277), seven newly identified STs and a novel allele related to *aroE* gene, demonstrating the high genetic diversity and an absence of relationship among most of our studied *P. aeruginosa* strains. Such findings are also in line with those obtained by rep-PCR genotyping, which revealed the presence of 36 fingerprints, 26 of which are unrelated (Figs 1 and 2). This high genetic diversity may be explained as the study site is a regional reference

hospital for several specialties. It is likely that inter-hospital transit patients and/or who spend short periods in the hospital environment, as well as healthcare professionals, carry isolates and insert them in the hospital environment, and consequently causing infections in ICU patients because of numerous risk factors, such as immunosuppression conditions and virulence potential of isolates. Our data also corroborate the hypothesis of epidemic non-clonal population structure of *P. aeruginosa*, indicating the dispersion of global HRC in hospitals settings and the continuous description of new *P. aeruginosa* STs worldwide [15, 23, 55, 56, 76, 85, 86].

Contrary to the predominance in other studies, HRC were associated with only 40% of isolates tested in this study [18, 23]. However, such strains demonstrate significantly association MDR/XDR phenotypes, carriage of resistance markers and persistence in ICUs (Table 5). ST111 has been detected in all continents [23, 31] and in our study was associated to clonal group I, composed by two MDR/XDR from different periods, of which only one harboring was $bla_{CTX-M1}$ gene, suggesting the latter acquisition of this marker (Fig 2). ST235 is the most widely spread *P. aeruginosa* HRC associated MDR/XDR isolates and carriage various resistance markers [23, 87]. Of the five ST235 isolates were MDR/XDR, of which three comprised the clonal group D, dispersed in different ICUs and carrying the gene $bla_{CTX-M1}$ gene, and two unrelated isolates, of which one also harboring $bla_{CTX-M1}$ gene (Fig 2). Kos et al. [88] has also detected the presence of CTX-M β-lactamase in Brazilian ST235 isolates. ST244 also has a worldwide distribution, being associated to carriage of different resistance markers, but not always related to MDR/XDR isolates [23, 88–90]. High frequencies were also reported by Brazilian studies evaluating clinical and environmental isolates [20, 76, 91]. ST244 was the most prevalent HRC among tested strains, being predominantly detected in AICU and mostly related to MDR/XDR isolates, except for one isolate presenting MultiS phenotype. Two of nine isolates comprised clonal group H/ST244, where $bla_{CTX-M1}$ was shared by both isolates and two comprised clonal group G/ST244; the other five isolates were genetically unrelated and three harbored $bla_{CTX-M1}$ gene (Fig 2). Endemic clone in Brazil, ST277 is frequently associated to MDR/XDR isolates and dissemination of $bla_{SPM-1}$ gene [20, 23, 92]. Interestingly, isolates comprising clonal group A/ST277 shared $bla_{SPM-1}$, but harbored distinct $bla_{OXA}$ genes; one isolate of clonal group B/ST277 harbored both $bla_{SPM-1}$ and $bla_{OXA-10}$ genes, while the other harbored only $bla_{OXA-10}$ gene; isolates comprising clonal group C/ST277 shared $bla_{SPM-1}$, but only one isolate harbored $bla_{OXA-2}$ (Fig 2). To our knowledge this is first report in Brazil of *P. aeruginosa* ST277 harboring both $bla_{SPM-1}$ and $bla_{OXA}$ variants genes.

Most of tested isolates were identified as non-HCR, which among them non-resistant phenotypes (MultiS and ModR) were significantly predominant ($P < 0.0001$) (Table 5), demonstrating that susceptible isolates are associated to higher clonal diversity [31]. Clonal group E/ST1284 comprised three isolates, which curiously, the only isolate harboring $bla_{CTX-M1}$ was negative for *exoU* gene, suggesting a reduction in virulence due to acquisition of a resistance marker (Fig 2). ST508 isolate was the only non-HRC harboring a resistance gene ($bla_{OXA-2}$) (Fig 2). Novel ST2554 was identified as DLV of HRC ST274, belonging to CC274. High-risk CC/ST274 strains have been recently detected in Indonesian ICUs, but are still predominant among CF patients, demonstrating high occurrence of mutation in genes related to antibiotic resistance [93–95]. ST2554 isolate presented a ModR phenotype and $exoS^+/exoU^+$ virulotype, also indicating acquisition of PAPI-2 containing *exoU* gene during divergence from ST274 (Fig 2). In spite of the non-detection of ST274 among isolates tested in this study, it may not rule out the presence of this genotype in the hospital, as we observe a more direct genetic relationship between both strains.

Certain clonal strains are expected to be linked to T3SS virulotypes. Recent phylogenetic analysis by Sawa et al. [19] attributed *exoU+* virulotype to ST253, possibly explaining the

highest virulence potential of ST235 observed clinical studies [18, 23]. Also, Sanchez-Diener et al. [72] on animal model-based study showed that lower virulence was linked to XDR profiles, which are typically found among HRC. However, different virulence potentials were observed among HRC, being higher for ST111 and ST235 and lower for ST175. Tested isolates MLST lineages were not significantly associated to specific T3SS virulotypes ($P = 0.5820$) (Table 5), however, we observed the presence of *exoS+*/*exoU+* virulotype among the four detected HRC. Moreover, we observed isolates HRC ST244 and non-HRC ST508 harboring *exoS+*/*exoU+* virulotype, while they were expected to be linked the invasive *exoS+* virulotype as recently reported [19]. Therefore, our findings highlight the importance of virulence genotyping and suggest the local acquisition of PAPI-2 containing *exoU* gene by HRC, which could worsen the local clinical scenario as most of HCR were also MDR/XDR.

The elevated virulence and resistance potential presented by isolates tested in this study might negatively affect patient's outcomes, however the present study did not perform associations to clinical and epidemiological data due to unavailability of patient's records. Moreover, we did not perform phenotypic tests evaluating secretion/expression of virulence factors such as elastases, proteases, T3SS exotoxins, QS molecules and the sequencing of the virulence genes, thus preventing a better establishment of genotype-phenotype relationships. Finally, the non-characterization of hospital environment isolates may have underestimated the genetic background in the study site, also impairing analysis of transmission from in-hospital environmental reservoirs. Studies comprising a larger number of clinical and environmental isolates from the study site and from other hospitals are highly recommended in order to better establish *P. aeruginosa* transmission in the region, as well as, relationships between of virulence, antibiotic susceptibility and genetic diversity.

## Conclusions

To conclude, this is the first study deeply exploring virulence, antibiotic resistance and genetic background of *P. aeruginosa* isolates from ICUs at specialized hospital in the Brazilian Amazon region. Herein, we reported a high prevalence of virulence-related markers, particularly of the rare *exoS+*/*exoU+* virulotype, which may be associated to a high virulence and cytotoxic potential and transmission of genetic elements containing *exoU* gene among local isolates. It was also observed high percentages of resistance to different classes of antibiotics, particularly to carbapenems, in addition to MDR/XDR isolates harboring resistance genetic elements, alerting to a possible dissemination of these mechanisms and of isolates acting as resistance reservoirs. The importance genotyping by rep-PCR and MLST techniques in the surveillance of in-hospital isolates was noteworthy, which demonstrated circulation of MDR/XDR HRC clones, emergence of novel genetic lineages and a wide genetic diversity of strains circulating in different ICUs. Finally, our findings may be useful on comprehension of complex mechanisms of pathogenicity process, treatment guidance and establishment of strategies to control the spread of epidemic *P. aeruginosa* strains with a high lethality potential.

## Supporting information

**S1 Table. Origin, resistance, genotyping and virulence data of the 54 *P. aeruginosa* isolates from the referral hospital under study.** * Novel ST. [a] Novel *aroE* allele. ICU, intensive care unit; AICU, adult intensive care unit; PICU, pediatric intensive care unit; NICU, neonatal intensive care unit; RTI, respiratory tract infection; BSI, bloodstream infection; FEP, cefepime; IMP, imipenem; MER, meropenem; TZP, piperacillin+tazobactam; GN, gentamicin; CIP, ciprofloxacin; POL, polymyxin B; MultiS, susceptible to all tested antibiotics; ModR, moderately

resistant; MDR, multi-drug resistant; XDR, extensively drug-resistant; ST, sequence type.
(XLSX)

**S1 Fig. Prevalence of virulence-associated genes in the 54 *P. aeruginosa* isolates from the referral hospital under study.**
(TIF)

## Acknowledgments

We thank Dra. Ana Roberta Fusco da Costa and Msc. Alex Brito Souza for the technical assistance in the MLST sequencing of samples.

The present study will be presented by Yan Corrêa Rodrigues to obtain his Ph.D. within the doctoral programme 'Parasitic biology in Amazon Region' at the State University of Pará, Brazil.

## Author Contributions

**Conceptualization:** Yan Corrêa Rodrigues, Karla Valéria Batista Lima.

**Formal analysis:** Yan Corrêa Rodrigues, Ismari Perini Furlaneto.

**Funding acquisition:** Karla Valéria Batista Lima.

**Investigation:** Yan Corrêa Rodrigues, Ismari Perini Furlaneto, Arthur Henrique Pinto Maciel, Ana Judith Pires Garcia Quaresma, Eliseth Costa Oliveira de Matos, Marília Lima Conceição, Marcelo Cleyton da Silva Vieira, Giulia Leão da Cunha Brabo, Edilene do Socorro Nascimento Falcão Sarges.

**Project administration:** Karla Valéria Batista Lima.

**Resources:** Luana Nepomuceno Godim Costa Lima.

**Supervision:** Yan Corrêa Rodrigues, Karla Valéria Batista Lima.

**Validation:** Yan Corrêa Rodrigues, Ismari Perini Furlaneto, Ana Judith Pires Garcia Quaresma, Karla Valéria Batista Lima.

**Writing – original draft:** Yan Corrêa Rodrigues, Ismari Perini Furlaneto, Luana Nepomuceno Godim Costa Lima, Karla Valéria Batista Lima.

**Writing – review & editing:** Yan Corrêa Rodrigues, Marcelo Cleyton da Silva Vieira, Luana Nepomuceno Godim Costa Lima, Karla Valéria Batista Lima.

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
