## [Decision Letter · Decision Letter 0]

28 Jul 2020

PONE-D-20-15488

High prevalence of atypical virulotype and genetically diverse background among Pseudomonas aeruginosa isolates from a referral hospital in the Brazilian Amazon

PLOS ONE

Dear Dr. RODRIGUES,

Thank you for submitting your manuscript to PLOS ONE. After careful consideration, we feel that it has merit but does not fully meet PLOS ONE’s publication criteria as it currently stands. Therefore, we invite you to submit a revised version of the manuscript that addresses the points raised during the review process.

We look forward to receiving your revised manuscript.

Kind regards,

Grzegorz Woźniakowski, Full professor, PhD, ScD

Academic Editor

PLOS ONE

Journal Requirements:

Reviewers' comments:

Reviewer's Responses to Questions

**Comments to the Author**

1. Is the manuscript technically sound, and do the data support the conclusions?

Reviewer #1: Yes

2. Has the statistical analysis been performed appropriately and rigorously? 

Reviewer #1: Yes

3. Have the authors made all data underlying the findings in their manuscript fully available?

Reviewer #1: Yes

4. Is the manuscript presented in an intelligible fashion and written in standard English?

Reviewer #1: Yes

5. Review Comments to the Author

Reviewer #1: The manuscript is technically sound and the presented data support the conclusions. Experiments have been conducted properly with reasonable sample size. The conclusions were drawn appropriately based on the data presented.There are some items need editing corrections as shown as sticky notes in the attached PDF file

6. PLOS authors have the option to publish the peer review history of their article (what does this mean?). If published, this will include your full peer review and any attached files.

Reviewer #1: **Yes: **Mohammad Aboulwafa

---

## [Author Response · Author response to Decision Letter 0]

3 Aug 2020

Journal Requirements:

Answer: Style requirements and file naming were corrected as requested. 

Review Comments to the Author: 

Reviewer #1: The manuscript is technically sound and the presented data support the conclusions. Experiments have been conducted properly with reasonable sample size. The conclusions were drawn appropriately based on the data presented. There are some items need editing corrections as shown as sticky notes in the attached PDF file.

Answer: We appreciate the reviewer comments. Editing corrections were incorporated through the manuscript and highlighted in the ‘Revised Manuscript with Track Changes’.

General Comment:

Fig 2 was corrected and improved as requested by Reviewer #1. In addition, Fig 1 was corrected and improved as requested by others co-authors.

---

## [Editor Report · Decision Letter 1]

24 Aug 2020

High prevalence of atypical virulotype and genetically diverse background among Pseudomonas aeruginosa isolates from a referral hospital in the Brazilian Amazon

PONE-D-20-15488R1

Dear Dr. RODRIGUES,

We’re pleased to inform you that your manuscript has been judged scientifically suitable for publication and will be formally accepted for publication once it meets all outstanding technical requirements.

Kind regards,

Grzegorz Woźniakowski, Full professor, PhD, ScD

Academic Editor

PLOS ONE
---

## [Editor Report · Acceptance letter]

26 Aug 2020

PONE-D-20-15488R1 

High prevalence of atypical virulotype and genetically diverse background among *Pseudomonas aeruginosa* isolates from a referral hospital in the Brazilian Amazon 

Dear Dr. RODRIGUES:

I'm pleased to inform you that your manuscript has been deemed suitable for publication in PLOS ONE. Congratulations! Your manuscript is now with our production department. 

Kind regards, 

on behalf of

Prof. Grzegorz Woźniakowski 

Academic Editor

PLOS ONE